# Transcriptomic Analysis of the CNL Gene Family in the Resistant Rice Cultivar IR28 in Response to *Ustilaginoidea virens* Infection

**DOI:** 10.3390/ijms251910655

**Published:** 2024-10-03

**Authors:** Zuo-Qian Wang, Yu-Fu Wang, Ting Xu, Xin-Yi Li, Shu Zhang, Xiang-Qian Chang, Xiao-Lin Yang, Shuai Meng, Liang Lv

**Affiliations:** 1Institute of Plant Protection and Soil Fertilizer, Hubei Academy of Agricultural Sciences, Wuhan 430064, China; 2Key Laboratory of Integrated Pest Management on Crops in Central China, Ministry of Agriculture, Wuhan 430064, China; 3Department of Plant Pathology, College of Plant Science and Technology, Huazhong Agricultural University, Wuhan 430070, China; 4State Key Laboratory of Subtropical Silviculture, Zhejiang A&F University, Hangzhou 311300, China

**Keywords:** *Ustilaginoidea virens*, NLR immune receptor, CNLs, transcriptome analysis

## Abstract

Rice false smut, caused by *Ustilaginoidea virens*, threatens rice production by reducing yields and contaminating grains with harmful ustiloxins. However, studies on resistance genes are scarce. In this study, the resistance level of IR28 (resistant cultivar) to *U. virens* was validated through artificial inoculation. Notably, a reactivation of resistance genes after transient down-regulation during the first 3 to 5 dpi was observed in IR28 compared to WX98 (susceptible cultivar). Cluster results of a principal component analysis and hierarchical cluster analysis of differentially expressed genes (DEGs) in the transcriptome exhibited longer expression patterns in the early infection phase of IR28, consistent with its sustained resistance response. Results of GO and KEGG enrichment analyses highlighted the suppression of immune pathways when the hyphae first invade stamen filaments at 5 dpi, but sustained up-regulated DEGs were linked to the ‘Plant–pathogen interaction’ (osa04626) pathway, notably disease-resistant protein RPM1 (K13457, CNLs, coil-coiled NLR). An analysis of CNLs identified 245 proteins containing Rx-CC and NB-ARC domains in the *Oryza sativa* Indica genome. Partial candidate CNLs were shown to exhibit up-regulation at both 1 and 5 dpi in IR28. This study provides insights into CNLs’ responses to *U. virens* in IR28, potentially informing resistance mechanisms and genetic breeding targets.

## 1. Introduction

Rice false smut, a disease caused by the ascomycete fungus *Ustilaginoidea virens* (Cooke) Takahashi (teleomorph: *Villosiclava virens*), has emerged as one of the most destructive phytopathogens affecting rice production [1,2]. Infection leads to significant yield reductions, and rice grains are also contaminated by ustiloxins of chlamydospores on the false smut balls [2]. Notably, rates of ustiloxin A (UA) and ustiloxin B detected on polished grain samples have, alarmingly, reached 82.1% and 49.3%. Furthermore, the rate of UA residues detected in urine samples was 22.8% in Enshi City, China, which may affect people’s health due to the induction of renal and hepatic impairments [3,4]. Thus, effective management strategies must be urgently developed to mitigate the negative impact of rice false smut on agriculture and public health.

Infection typically occurs during the booting stage before pollen matures, and hyphae of *U. virens* infiltrate the spikelet’s interior through the gaps between the lemma and palea [5]. The pathogen then invades the intercellular spaces of stamen filaments and ovaries, usurping nutrients for mycelial growth by mimicking the process of ovary fertilization [1]. This process of infection ends with the formation of a false smut ball laden with toxic chlamydospores. No lesions were observed, and the ovary was fresh during infection, suggesting a suppression of the immune system, potentially through the deployment of effector proteins [6,7]. Although planting resistant rice cultivars is a cost-effective and eco-friendly approach to disease control, the gene resources for *U. virens* resistance breeding remain lacking [8].

Genetic engineering strategies have shown benefits for disease-resistance breeding, such as the targeted editing of lesion mimic mutant (LMM) genes [9]. Similarly, an overexpression of the rice scaffold protein OsRACK1A, a target of the effector UvCBP1, confers resistance to *U. virens* without adversely affecting yield [10]. Additionally, editing the OsSRT2 protein, the target of effector Uv1809, has demonstrated broad-spectrum resistance to rice diseases, including *U. virens* [11]. These findings suggest that the genetic manipulation of resistance-related genes could provide valuable breeding resources [12]. Although qualitative resistance genes specific to *U. virens* have not yet been identified, candidate resistance-related quantitative trait loci (QTLs) have been mapped in resistant rice cultivars and varieties [13,14,15]. For example, the QTL *qRFS12.01* on chromosome 12 has been identified using bulked segregant analysis (BSA) and simple sequence repeat (SSR) marker mapping in the resistant variety IR77298-14-1-2::IRGC117374-1, revealing five nucleotide-binding leucine-rich repeat (NLR) domain-containing proteins [16]. Furthermore, a novel QTL *qRFSr9.1* on chromosome 9 in the resistant breeding line RYT2668 has also been associated with four NLR domain-containing proteins [17]. Thus, NLR genes may be a potential target for genetically engineering resistance due to their relation to resistance against *U. virens*.

NLR genes are essential for the plant’s immune response to pathogens as intracellular immune receptors, enabling the recognition of pathogen effectors and the initiation of effector-triggered immunity or broad-spectrum resistance [18,19]. In plants, NLR genes are divided into CNLs (contain a coiled-coil domain), TNLs (contain a Toll/interleukin-1 receptor (TIR) domain), and RNLs (contain an N-terminal RPW8 domain) [20,21]. CNLs have been found exclusively in monocots that can initiate the formation of a resistosome. This large hetero-oligomeric protein complex functions as a calcium channel and is central to CNL-mediated immunity [22]. In the case of rice blasts caused by *Magnaporthe oryzae*, over 100 R genes have been mapped, and the majority of the 31 cloned R genes encode CNLs [23], such as Pid3, Pit, Pi25, etc. [24]. For instance, Pid3 mediates disease resistance by activating RAC Immunity1(RAI1) through the OsSPK1 signaling pathway [25]. In the case of rice bacterial blight, most R genes also encode CNLs [26]. However, the role of rice CNLs in response to *U. virens* has not been reported.

The aim of this study was to unveil resistant mechanisms and find resistant-related gene resources from the resistant cultivar IR28. In this study, the resistance levels of IR28 and susceptible cultivar WX98 were evaluated with artificial inoculation pathogenicity analysis and a quantification of resistance genes OsBETV1 and OsPR10b. Then, a comparative transcriptomic analysis was performed to evaluate the expression profiles of DEGs between IR28 and WX98. Gene Ontology (GO) enrichment and Kyoto Encyclopedia of Genes and Genomes (KEGG) pathway analysis were employed to identify biological processes and pathways associated with resistance. CNL genes within the *Oryza sativa* indica genome were systematically screened for the presence of an NB-ARC domain and a coiled-coil domain. This approach enabled the identification of candidate CNLs that are responsive to *U. virens* infection. The differential expression of these candidate genes was validated using quantitative real-time polymerase chain reactions (qRT-PCRs). The transcriptomic-level understanding of CNL’s family response to *U. virens* infection not only advances knowledge on the mechanism of the resistance to *U. virens* in rice but also provides a foundation for discovering resistance breeding resources.

## 2. Results

### 2.1. Disease Symptoms of U. virens-Inoculated IR28 and WX98 Cultivars

The resistance phenotypes of the rice cultivars IR28 and WX98 were assessed following inoculation with two *U. virens* isolates. Figure 1A displays the panicles and corresponding false smut balls from each cultivar, revealing significantly lower average numbers in the resistant IR28 (6.65 ± 8.13 and 6.60 ± 6.31) compared to the susceptible WX98 (38.70 ± 24.40 and 30.60 ± 24.96) when inoculated with isolates HWD-2 and JS60-2, respectively (*p* < 0.001). The percentage of infected flowers in IR28 (5.54 ± 6.58 and 5.54 ± 5.33) was also significantly lower than in WX98 (23.10 ± 13.76 and 16.53 ± 10.98), as detailed in Figure 1B and Table 1.

The temporal expression profiles of resistance-related genes OsBETV1 and OsPR10b were analyzed via qRT-PCR following conidia injection inoculation at 0, 1, 3, 5, and 7 days post inoculation (dpi) in both cultivars. In the resistant cultivar IR28, both genes exhibited an initial up-regulation at 1 dpi, a transient down-regulation at 3 dpi, and a pronounced resurgence at 5 and 7 dpi. This pattern suggests a transient suppression of the immune response in IR28 at 3 dpi, which was subsequently reactivated by 5 dpi. Specifically, the expression level of OsBETV1 in IR28 was significantly up-regulated by 1.53-fold and 10.75-fold compared to WX98 at 5 dpi and 7 dpi, respectively. The expression level of OsPR10b in IR28 was significantly up-regulated by 3.03-fold and 21.42-fold relative to WX98 at 1 dpi and 7 dpi, respectively, as illustrated in Figure 1C.

To further elucidate the mechanisms of the resistance response to *U. virens* in IR28, the expression levels of the resistance genes were assessed in panicles (infection site), leaves, and stems at 3 dpi and 5 dpi. The analysis was conducted under both inoculated and non-inoculated conditions. The results exhibited a significant up-regulation of the two resistance genes in inoculated panicles and leaves compared to non-inoculated samples. This induction was observed at both 3 and 5 dpi, indicating that the inoculation with *U. virens* effectively triggered the expression of resistance genes in the panicle of IR28. Interestingly, the up-regulated expression of resistance genes was observed in leaves that were not within the infection site, suggesting that the resistance response in IR28 is not limited to local defense but involves a systemic immune response.

### 2.2. Transcriptome Profiling and Identification of Differentially Expressed Genes

RNA-seq was deployed to characterize the resistance gene network and its regulatory response infection in the resistant cultivar IR28 in contrast to susceptible cultivar WX98 following *U. virens* inoculation. Panicle samples from both cultivars were collected at 1, 5, 9, and 13 dpi, with three independent biological replicates. The RNA-seq analysis generated 166.55 gigabases (Gb) of pair-end raw reads and 163.15 Gb of clean reads after quality filtering across the 24 samples. The average Q20 value of the reads was 96.77%, and the average Q30 value was 92.18%. These reads were mapped to the reference genome of the *Oryza sativa* Indica group (ASM465v1) with an average mapping ratio of 84.10%. On average, 28,648.54 genes were mapped per sample (Appendix A). The statistics for the RNA-seq data indicate a reliable and comprehensive dataset for the subsequent analysis of differential gene expression and regulatory pathways.

Pearson correlation coefficients (PCCs) were calculated based on the FPKM (Fragments Per Kilobase of transcript per Million mapped reads) values of expressed genes across twelve samples at all four dpi of each cultivar. The analysis of these correlation coefficients provided evidence of the reproducibility and reliability of the gene expression data for the biological replicates for both cultivars, as illustrated in Appendix A.

Principal component analysis (PCA) was performed to assess the variance in gene expression dynamics between the resistant cultivar IR28 and the susceptible cultivar WX98 under *U. virens* infection. The PCA results revealed a clear separation between the samples of the two cultivars, indicating significant differences in their expression profiles (Figure 2A). Within the resistant cultivar IR28, samples clustered into two distinct groups based on the post-inoculation time points: one group consisting of samples from 1, 5, and 9 dpi, and another group containing samples from 13 dpi. In contrast, for the susceptible cultivar WX98, samples from 1 and 5 dpi were in one group, while those from 9 and 13 dpi constituted another. These observations suggest the presence of two distinct phases of gene expression dynamics during the infection process, with the timing of these phases differing between the two cultivars. The early phase, potentially related to the resistance of IR28, appears to be longer than that of WX98.

The results of the hierarchical cluster analysis (HCA) were consistent with those of the PCA; 24 samples were separated between two cultivars and divided into two phases and five subgroups, indicating different expressions of the rice cultivars IR28 and WX98 following *U. virens* inoculation. Subgroup I included three samples at 13 dpi, indicating a transcriptional state at the late infection phase in IR28. Subgroup II consisted of WX98 samples from 9 and 13 dpi, suggesting infection progressed faster in WX98. Subgroups III, IV, and V contained samples from both cultivars at the early phase post inoculation: Subgroup III comprised WX98 samples at 1 and 5 dpi; Subgroup IV included IR28 samples at 1 dpi; and Subgroup V encompassed IR28 samples at 5 and 9 dpi. Comparing Subgroup III with Subgroups IV and V highlighted the sustained transcriptional response in the infiltered spikelet of IR28 by *U. virens* at the early phase (Figure 2B).

A differential gene expression analysis was conducted to identify genes with significant changes in response to *U. virens* infection, using a threshold of log^2^ fold change > 1 and an adjusted *p*-value of < 0.05. At 1 dpi, 8288 differentially expressed genes (DEGs) were identified between IR28 and WX98, with 3943 down-regulated genes (most significant change: −14.45 log^2^ fold change, adjusted *p*-value: 7.84 × 10^−33^) and 4345 up-regulated genes (highest change: 14.66 log^2^ fold change, adjusted *p*-value: 8.64 × 10^−34^). At 5 dpi, 7709 DEGs were detected, with 3570 down-regulated (minimum: −14.02 log^2^ fold change, adjusted *p*-value: 6.98 × 10^−30^) and 4139 up-regulated (maximum: 14.22 log^2^ fold change, adjusted *p*-value: 2.39 × 10^−30^). At 9 dpi, the number of DEGs increased to 11,366, with 6018 being down-regulated (minimum: −17.86 log^2^ fold change, adjusted *p*-value: 4.24 × 1^−18^) and 5318 up-regulated (maximum: 14.46 log^2^ fold change, adjusted *p*-value: 1.09 × 1^−23^). At 13 dpi, 7700 DEGs were identified, comprising 3169 down-regulated genes (minimum: −14.44 log^2^ fold change, adjusted *p*-value: 5.79 × 10^−32^) and 4531 up-regulated genes (maximum: 13.90 log^2^ fold change, adjusted *p*-value: 4.17 × 1^−30^) (Figure 3A,B). The highest number of DEGs was at 9 dpi when the hyphae reached the stigma and ovary [27], suggesting a potent transcript response to infection in IR28.

### 2.3. GO Enrichment Analysis of Differentially Expressed Genes (DEGs)

To identify DEGs and associated pathways related to resistance mechanisms in response to *U. virens* infection globally, a GO enrichment analysis was performed on 2585 DEGs that exhibited differential expression in IR28 compared to WX98 at 1, 5, 9, and 13 dpi, using an adjusted *p*-value < 0.05 (Figure 3C). The most significantly enriched GO term was “ionotropic glutamate receptor activity” (GO:0004970). Other enriched terms included “extracellular ligand-gated ion channel activity” (GO:0005230), “glutamate receptor activity” (GO:0008066), and a series of related GO terms reflecting the involvement of neurotransmitters and ligand-gated ion channels, such as transmitter-gated ion channel activity (GO:0022824) and ligand-gated ion channel activity (GO:0015276). Additionally, terms associated with transmembrane signaling and ion channel activities, such as transmembrane signaling receptor activity (GO:0004888) and ion channel activity (GO:0005216), were also enriched (Figure 3D). The enrichment of these GO terms suggests that a variety of ion channel activities and neurotransmitter receptor functions are up-regulated in IR28, indicating a potent immune signal transportation occurs in IR28.

At 1 dpi, three of the ten enriched GO terms belonged to the molecular function (MF) category, including “Peroxidase activity” (GO:0004601), “Oxidoreductase activity, acting on peroxide as acceptor” (GO:0016684), and “Antioxidant activity” (GO:0016209). The remaining terms were categorized in the biological process (BP), with “Cellular oxidant detoxification” (GO:0098869), “Detoxification” (GO:0098754), and “Cellular detoxification” (GO:1990748) being the most prominent (Figure 3E). By 5 dpi, the most enriched terms had shifted to cellular component (CC) terms, with “Thylakoid” (GO:0009579), “Thylakoid part” (GO:0044436), and “Photosystem” (GO:0009521) being the top three, with the rest of the top ten all related to chloroplasts and photosynthesis (Figure 3F). At 9 dpi, the enriched terms were evenly distributed across the BP, CC, and MF categories. Key terms included “Drug catabolic process” (GO:0042737) and “Cell wall organization or biogenesis” (GO:0071554) in BP, “Cell wall” (GO:0005618) and “External encapsulating structure” (GO:0030312) in CC, and “Hydrolase activity, hydrolyzing O-glycosyl compounds” (GO:0004553), “Active transmembrane transporter activity” (GO:0022804), and “Secondary active transmembrane transporter activity” (GO:0015291) in MF (Figure 3G). At 13 dpi, nine out of the top ten enriched terms were CC-related, with “Photosynthetic membrane” (GO:0034357), “Thylakoid part” (GO:0044436), and “Photosystem” (GO:0009521) leading, and “Photosynthesis” (GO:0015979) being the sole BP term (Figure 3H). The fact that enriched terms at 1 and 9 dpi were related to immune response but at 5 and 13 dpi were related to plant development suggest a robust immune response at 1 and 9 dpi, with a suppression of immune activity at 5 dpi.

### 2.4. KEGG Pathway Enrichment Analysis of DEGs in Plant–Pathogen Interactions

To elucidate the distinct biological processes activated in the rice cultivars IR28 and WX98 in response to *U. virens* infection, a KEGG pathway enrichment analysis was conducted on differentially expressed genes (DEGs) at 1, 5, 9, and 13 dpi separately. At 1 dpi, the up-regulated DEGs in IR28 were significantly enriched (adjusted *p*-value < 0.05) in pathways “Phenylpropanoid biosynthesis” (osa00940), “Plant-pathogen interaction” (osa04626), and the “MAPK signaling pathway” (osa04016). These pathways are critical for the biosynthesis of defense-related compounds and signal transduction mechanisms [28,29]. At 5 dpi, enrichment was most pronounced in pathways associated with “Photosynthesis-antenna proteins” (osa00196) and “Photosynthesis” (osa00195), which were related to panicle development and grain filling. Pathways related to plant defense, such as “Flavonoid biosynthesis” (osa00941) and “Phenylpropanoid biosynthesis” (osa00940), were also significantly enriched [30]. At 9 dpi, the enrichment profile included pathways involved in “Phenylpropanoid biosynthesis” (osa00940), “Pentose and glucuronate interconversions” (osa00040), and “alpha-Linolenic acid metabolism” (osa00592), which are consistent with results of a study in which infection of *U. virens* in WX98’s spikelets stopped pollen maturity and grain filling [1]. Except for these grain-filling-related pathways, continued enrichment in the “Plant–pathogen interaction pathway” (osa04626) suggests a sustained host–pathogen interaction in IR28. At 13 dpi, the up-regulated DEGs were significantly enriched in pathways related to energy production and photosynthesis, such as “Photosynthesis-antenna proteins” (osa00196) and “Photosynthesis” (osa00195), indicating that the interaction between *U. virens* and the plants has ended, consistent with hypha growing and embracing all the inner floral parts at 13 dpi [27]. Similar to GO enrichment analysis, resistance-related pathways were up-regulated at 1 dpi and 9 dpi, but their expression may be suppressed at 5 dpi. Notably, although the number of DEGs enriched in the resistance-related pathways was reduced at 5 dpi, there were still DEGs enriched in pathways like “Flavonoid biosynthesis” (osa00941) (Appendix A). This finding suggests that DEGs active at 5 dpi could be crucial for resistance to *U. virens*.

A total of 2404 DEGs were significantly up-regulated in IR28 compared to WX98 at both 1 and 5 dpi (adjusted *p*-value < 0.05). KEGG pathway enrichment analysis revealed significant enrichment in “Phenylpropanoid biosynthesis” (osa00940), “Flavonoid biosynthesis” (osa00941), and “Plant–pathogen interaction” (osa04626) (*p* < 0.05) (Figure 4A and Table 2). Within the plant–pathogen interaction pathway, 26 out of 302 enriched DEGs were directly related to resistance. Consequently, DEGs enriched in the plant–pathogen interaction pathway were selected for heatmap construction (Figure 4B). In this pathway, BGIOSGA036090 and BGIOSGA015563, enriched in RPM1 (disease resistance protein RPM1, K13457), contain NB-ARC and Rx-CC domains and are described as NB-LRR disease resistance proteins, showing an up-regulation of 24.84- to 29.88-fold and 25.58- to 29.51-fold at 1 and 5 dpi, respectively. BGIOSGA015767 and BGISGA031149, enriched in HSP90 (heat shock protein 90 kDa beta, K09487), contain HSP90 domains and were up-regulated 21.87- to 27.51-fold at 1 dpi and 21.94- to 27.74-fold at 5 dpi. BGIOSGA008987 and BGIOSGA010624, enriched in KCS1/10 (3-ketoacyl-CoA synthase, K15397), contain CHS domains, described as type III polyketide synthase, and were up-regulated 23.01- to 23.72-fold at 1 dpi and 23.35- to 24.85-fold at 5 dpi. BGIOSGA000109, BGIOSGA018153, BGIOSGA018556, BGIOSGA000087, BGIOSGA002446, BGIOSGA023203, BGIOSGA017499, and BGIOSGA005163, enriched in CaMCML (calcium-binding protein CML, K13448), contain PTZ00183, PTZ00184, and Efh domains, described as EF-hand proteins, and were up-regulated 21.13- to 29.29-fold at 1 dpi and 21.10- to 27.26-fold at 5 dpi. BGIOSGA014554, enriched in CDPK (calcium-dependent protein kinase, K13412), contains PKc_like domains and was up-regulated 21.76-fold at 1 dpi and 22.04-fold at 5 dpi. BGIOSGA020131, enriched in CNGCs (cyclic nucleotide-gated channel, K05391), contains CAP_ED domains, described as cyclic nucleotide-binding proteins, and was up-regulated 24.98-fold at 1 dpi and 25.02-fold at 5 dpi. BGIOSGA010959, enriched in MPK3/6 (mitogen-activated protein kinase 3, K20536), contains PKc_like domains and was up-regulated 22.23-fold at 1 dpi and 21.01-fold at 5 dpi. BGIOSGA001599, BGIOSGA025088, BGIOSGA025078, and BGIOSGA025083, enriched in PR1 (pathogenesis-related protein 1, K13449), contain CAP domains and were up-regulated 21.95- to 211.32-fold at 1 dpi and 21.51- to 212.56-fold at 5 dpi. BGIOSGA001663, enriched in Rboh (respiratory burst oxidase, K13447), contains NADPH_Ox, FNR_like, and FAD_binding_1 domain, described as respiratory burst oxidase homolog proteins, and was up-regulated 21.10-fold at 1 dpi and 21.14-fold at 5 dpi. BGIOSGA020451, BGIOSGA001178, and BGIOSGA011802, enriched in WRKY22 (WRKY transcription factor 22, K13425), contain WRKY and PLN02192 domains, described as WRKY transcription factors, and were up-regulated 21.54- to 28.59-fold at 1 dpi and 21.30- to 28.69-fold at 5 dpi. BGIOSGA030067, enriched in WRKY33 (WRKY transcription factor 33, K13424), contains WRKY domains and was up-regulated 22.68-fold at 1 dpi and 23.35-fold at 5 dpi (Figure 4C and Table 3).

### 2.5. Analysis of CNL Genes in Rice in Response to U. virens Infection

RPM1-like NLR genes are integral to the frontline of immune receptors for pathogen recognition. They exhibit a pivotal response to *U. virens* infection, particularly during the early stages of 1 to 5 dpi. To identify proteins in the rice genome containing Rx_CC (Coiled-coil) and NB-ARC domains, a Hidden Markov Model profile for Rx_CC was trained with reviewed protein sequences in the conserved domain collection CD14798 as a training dataset https://www.ebi.ac.uk/interpro/entry/cdd/CD14798/protein/reviewed/ (acceseed on 21 April 2023).

The results of an hmmsearch (https://www.ebi.ac.uk/Tools/hmmer/search/hmmsearch (acceseed on 21 April 2023)) identified 295 proteins containing the Rx_CC domain and 383 proteins with the NB-ARC domain. A total of 245 proteins with both Rx_CC and NB-ARC domains were identified in the *Oryza sativa* Indica genome (Figure 5A). These CNL genes were distributed across all 12 chromosomes, with varying numbers identified on each chromosome: 22 on chromosome I, 19 on chromosome II, 12 on chromosome III, 19 on chromosome IV, 15 on chromosome V, 22 on chromosome VI, 9 on chromosome VII, 25 on chromosome VIII, 12 on chromosome IX, 14 on chromosome X, 46 on chromosome XI, and 25 on chromosome XII (Figure 5B). The duplication analysis classified the duplicated genes into 22 segmental, 103 tandem, 30 proximal, and 90 dispersed duplicates (Figure 5B).

The transcriptome profiles of 113 genes were analyzed and grouped into six clusters based on a heatmap. Group I contained 16 NLRs that are up-regulated at 1 dpi in the IR28 cultivar. Group II comprised seven NLRs up-regulated at 5 dpi in IR28. Group III comprised 24 NLRs up-regulated at 1 and 5 dpi in IR28 compared to WX98. Group IV consisted of 22 NLRs significantly up-regulated in WX98 at 5 dpi. Group V comprised 10 NLRs up-regulated in WX98 at 1 dpi, and Group VI contained 23 NLRs up-regulated in WX98 at 1 and 5 dpi. These results suggest that the NLRs in Groups I, II, and III may be associated with resistance to *U. virens* in the IR28 cultivar (Figure 5C).

### 2.6. Validation of Candidate Genes Via qRT-PCR

An analysis of the expression dynamics of twelve candidate CNL genes following *U. virens* inoculation was conducted. These genes, ranging from 1080 bp to 3993 bp in length, are distributed across chromosomes 1, 8, 9, 10, 11, and 12. Subcellular localization predictions, based on their amino acid sequences and utilizing the WoLF PSORT (https://wolfpsort.hgc.jp/, 21 April 2023), indicated that BGIOSGA003230, BGIOSGA003329, BGIOSGA029433, BGIOSGA032859, and BGIOSGA036090 are localized to the cytoplasm, BGIOSGA028077 and BGIOSGA032258 to the integral membrane, BGIOSGA032326, BGIOSGA033963, and BGI-OSGA035581 to the nucleus, and BGIOSGA035675 and BGIOSGA036720 to the chloroplast and mitochondria (Table 4). The expression of these genes was monitored at 0, 1, 3, 5, and 7 dpi in two rice cultivars. Notably, BGIOSGA003230, BGIOSGA003329, BGIOSGA028077, BGI-OSGA032258, BGIOSGA032326, BGIOSGA033963, BGIOSGA036090, and BGIOSGA036720 demonstrated significant up-regulation in the resistant IR28 cultivar, compared to the susceptible WX98, from 0 to 7 dpi, confirming their induced expression in response to *U. virens* infection (Figure 6).

The expression levels of candidate CNLs were also evaluated in the panicle, leaf, and stem of IR28 following *U. virens* inoculation, with non-inoculated plants as controls. Notably, BGIOSGA003230, BGIOSGA003329, BGIOSGA028077, BGIOSGA029433, and BGIOSGA032859 exhibited up-regulation in the panicle at 3 dpi. Meanwhile, BGIOSGA033963, BGIOSGA030690, and BGIOSGA036720 showed increased expression in the panicle at both 3 and 5 dpi, suggesting a sustained induction during the infection process. Except for BGIOSGA029433 and BGIOSGA035581, these genes displayed up-regulated expression in both leaf and stem tissues, indicating a signal transduction mechanism from the panicle initiating a systemic response (Figure 7). These results provide insights into the time-dependent and tissue-specific expression patterns of the candidate CNL genes in IR28, highlighting their potential roles in the resistance mechanism against *U. virens* infection.

## 3. Discussion

Our study demonstrated a clear difference in resistance to *U. virens* between the rice cultivars IR28 and WX98. The significantly lower incidence of false smut in IR28 exhibits the effectiveness of resistant cultivars in disease management. The temporal gene expression analysis of OsBETV1 and OsPR10b in IR28 shows an initial immune response, a subsequent suppression period, and a later resurgence. This “N-shaped pattern” suggests initial pathogen recognition followed by subsequent reactivation of the host immune system in response to the latent invasion of the pathogen in IR28. These findings highlight the importance of understanding the molecular basis of IR28’s resistance.

Transcriptomic profiling also revealed a special gene expression response to *U. virens* infection in IR28. High-quality RNA-seq data mapped to the *Oryza sativa* Indica genome can form reliable datasets for differential gene expression analysis. PCA and HCA revealed distinct gene expression patterns between the resistant IR28 and susceptible WX98 cultivars, which indicates that IR28 shows a durable response in initial defense and subsequent robust resistance at the transcriptome level. Changes in the number of DEGs identified at various post-inoculation time points also highlighted differences in pathogen–plant interactions during the infection period. The results observed pathologically were consistent with the following infection process: at 7 dpi, the hyphae encircle the filaments and stamens, and at 9 dpi, the hyphae fill the lower part of the spikelet, indicating successful infection of its internal organs [1,27]. Thus, the interaction with the pathogen and the immune response may be sustained beyond 5 dpi to stop the infection in IR28.

The results of the GO enrichment analysis and KEGG pathway analysis provide strong evidence that IR28 is more resistant than WX98. DEGs significantly enriched in plant-defense-signal-related GO terms in IR28 confirmed the resistance to *U. virens* at the transcriptomic level. At the same time, the fact that DEGs were enriched in panicle-development and grain-filling-related pathways also revealed that those processes were intercepted by *U. virens* in WX98 at 5, 9, and 13 dpi at the KEGG pathway level. More importantly, the GO enrichment analysis of DEGs revealed “double top” expression alteration that up-regulated DEGs, enriched in defense-related GO function at 1 and 9 dpi, were suppressed at 5 dpi. Additionally, the significant up-regulation of DEGs enriched in KEGG pathways is consistent with the results of GO enrichment analysis; they were enriched in defense-related pathways at 1 dpi, suppressed at 5 dpi, and showed a resurgence at 9 dpi. After 5 dpi, the “war” between the host and pathogen shifted from confrontation to “Trojan horse”-like behind-enemy-lines infiltration, with the pathogen attempting to deceive the immune system to complete the infection. Most of the enrichment pathways were not plant-defense-related at 5 dpi, which may indicate that the immune system was cheated and suppressed, whereas the minor enriched DEGs in the plant–pathogen interaction pathway in IR28 suggest an underground battle. Thus, the specific DEGs up-regulated at 5 dpi, such as those encoding CNL receptors, may still be functional for pathogen recognition.

In total, 245 CNLs were identified with Rx-CC and NB-ARC domains across the *Oryza sativa* Indica genome; they were clustered into six distinct groups based on their expression at 1 dpi and 5 dpi. The up-regulated CNLs in Groups I, II, and III were considered to be resistance-related genes in IR28. The qRT-PCR validation of candidate CNLs has provided strong evidence of their expression in response to *U. virens* infection in IR28. Notably, the tissue-specific and time-dependent expression patterns observed for genes such as BGIOSGA033963, BGIOSGA030690, and BGIOSGA036720, which were induced in panicles at both 3 and 5 dpi, suggest a high correlation between CNLs and *U. virens* resistance. Therefore, investigating the function of CNLs may help unveil resistance mechanisms and provide a target for genetic engineering breeding against *U. virens*.

## 4. Materials and Methods

### 4.1. Plant Material and Fungal Strains and U. virens Inoculation

Rice *Oryza sativa* cvs. “IR28” and “Wanxian98 (WX98)” were used in this study. IR28 was cultivated by the International Rice Research Institute in Manila, Philippines, with the cross of IR833-6-1-1-1/IR1561-149-1 as the maternal parent and IR1737 as the paternal parent. WX98 was developed by the Hubei Academy of Agricultural Sciences in Wuhan, China, with “D0424S” as the female parent and “Minghui 63” as the male parent. Both varieties are of the indica rice type. They were planted in pots and moved into a glasshouse when inoculated. The *U. virens* strains HWD-2 and JS60-2 were used for pathogenicity assays. Panicles of rice plants at the booting stage were injected and inoculated with a mycelia and spore suspension (1 × 10^7^ conidia/mL) (number of panicles = 20), and the number of false smut balls and infected flower ratio were counted after three weeks [31]. The inoculation experiments were conducted in a greenhouse at a temperature of 28 °C with a humidity level exceeding 90%. All pathogenicity experiments were repeated twice. The pathogenicity analysis was performed in 2020 and 2021.

### 4.2. RNA-Seq of IR28 and WX98 Following U. virens Inoculation

Panicles from rice cultivars IR28 and WX98 were collected at 1, 5, 9, and 13 dpi with *U. virens* strain HWD-2, with three biological replicates for each time point, resulting in 24 total libraries. Total RNA was extracted using Trizol (Invitrogen, Waltham, MA, USA), and its integrity was verified using the RNA Nano 6000 Assay Kit on a Bioanalyzer 2100 system (Agilent Technologies, Santa Clara, CA, USA). mRNA was isolated using poly T oligo-attached magnetic beads and fragmented with divalent cations. First-strand cDNA synthesis was performed with random hexamers and M-MuLV Reverse Transcriptase, followed by second-strand synthesis using DNA Polymerase I and RNase H. The resulting cDNA fragments were blunt ended, adenylated at the 3′ ends, and ligated with adaptors. Libraries containing cDNA fragments of 370–420 bp were purified using the AMPure XP (Beckman Coulter, Indianapolis, IN, USA) and amplified by PCR (Bio-Rad, Hercules, CA, USA) with Phusion High-Fidelity DNA polymerase. Library quality was assessed on a Bioanalyzer 2100 system.

Raw reads were processed to remove adapters, poly N sequences, and low-quality reads, yielding high-quality clean reads for downstream analyses. Quality metrics such as Q20, Q30, and GC content were calculated. Clean reads were aligned to the *Oryza sativa* Indica reference genome (ASM465v1) using Hisat2 v2.0.5, chosen for its ability to handle splice junctions and provide superior mapping results.

### 4.3. Differentially Expressed Genes, GO Enrichment, and KEGG Enrichment

Reads were quantified using FeatureCounts v1.5.0-p3, and FPKM values were calculated to account for gene length and sequencing depth, providing normalized expression levels. Differential expression analysis between conditions (using two biological replicates each) was conducted with DESeq2 v1.20.0, which employs a negative binomial model. *p*-values were adjusted for multiple testing using the Benjamini–Hochberg method, with genes having adjusted *p*-values ≤ 0.05 considered differentially expressed. Pearson correlation coefficients of 24 samples were calculated in R studio to evaluate the repeatability.

Gene Ontology (GO) enrichment analysis was performed using the clusterProfiler R package, correcting for gene length bias, and GO terms with corrected *p*-values < 0.05 were deemed significantly enriched. KEGG pathway enrichment analysis was also conducted using clusterProfiler to understand the biological functions and utilities from large-scale molecular datasets.

### 4.4. qRT-PCR Evaluation of Resistance-Related Genes and Candidate CNLs

OsBETV1 (LOC_Os12g36850) and OsPR10b (AAF85973.1) were used to evaluate resistance against *U. virens* in some studies [10,32]. To evaluate the expression of the resistance-related genes OsBETV1 and OSPR10b during *U. virens* infection, panicles from rice cultivars IR28 and WX98 inoculated with *U. virens* HWD-2 were collected at 0, 1, 3, 5, and 7 dpi. Additionally, both inoculated and non-inoculated panicles and leaves on the same plant from IR28 were collected. For the validation of candidate CNLs using qRT-PCR (Bio-Rad), panicles from IR28 and WX98 were collected at 0, 1, 3, 5, and 7 dpi. In IR28, inoculated and non-inoculated samples of panicles, leaves, and stems were also collected at 3 and 5 dpi. RNA was extracted using the MolPure^®^ Plant RNA Kit (Yeasen, Shanghai, China) and reverse-transcribed into cDNA using Hifair^®^ III Reverse Transcriptase with oligo (dT) 18 primers (Yeasen, Shanghai, China). The expression levels of the resistance-related genes and candidate CNLs were assessed using qRT-PCR on an Applied Biosystems StepOnePlus™ Real-Time PCR System with Hieff^®^ qPCR SYBR Green Master Mix (Yeasen, Shanghai, China) in 10 μL reaction volumes. All qRT-PCR experiments were performed with three biological replicates. Primers used in this study are listed in Appendix A.

### 4.5. Principal Component Analysis (PCA) and Heatmap

Transcripts with an abundance greater than 1 FPKM were selected for the calculation of pairwise correlations among samples. Pairwise correlation matrices for the 24 samples to *U. virens* infection at 1, 5, 9, and 13 dpi were generated. A heatmap was built using the R package ‘pheatmap’. Samples were clustered based on FPKM values of DEGs, with the ‘cutree’ function within pheatmap, and distinct clusters were identified and separated according to the KEGG enrichment node in “Plant–pathogen interaction” (osa04626) pathways. For the CNL expression, cluster heatmaps were built based on the FPKM values of 245 CNLs.

### 4.6. Identification of CNL Gene Family Members

Genome-wide identification of CNL genes from *Oryza sativa* L. ssp. Indica was performed using Hidden Markov Model (HMM) version 3.0 with CC and NB-ARC HMM profiles. The threshold was set at an e-value ≤ 1e^−20^ [33]. The CC HMM profile was trained using a collection of coiled-coil conserved protein sequences (cd14798) obtained from InterPro https://www.ebi.ac.uk/interpro/entry/cdd/CD14798/protein/reviewed/ (accessed on 6 July 2023). The profile was built with the command hmmbuild based on an alignment of the N-terminal sequences. The NB-ARC HMM profile was obtained from the Pfam database in InterPro with accession number IPR002182 (https://www.ebi.ac.uk/interpro/entry/InterPro/IPR002182/ (accessed on 11 August 2023). Results containing both CC and NB-ARC motifs were selected. Conserved domains of the proteins were identified using the NCBI Batch CD-search tool https://www.ncbi.nlm.nih.gov/Structure/bwrpsb/bwrpsb.cgi (accessed on 9 September 2023). Localization predictions were performed with WoLF PSORT https://wolfpsort.hgc.jp/ (accessed on 18 October 2023).

### 4.7. Gene Structure Analysis, Chromosomal Distribution, and Gene Duplication

Gene duplication was identified using MCScanX with a replicate gene classifier program [34]. All amino acid sequences of all predicted NLR proteins were aligned using blastp, and alignment results with corresponding parsed GFF (General Feature Format) files were used as input. Output results classify the duplicate genes of a single species into WGD/segmental, tandem, proximal, and dispersed duplicates [34].

### 4.8. Statistical Analyses

Statistical significance among the datasets was determined using one-way analysis of variance (ANOVA) with Duncan’s Multiple Range Test. These analyses were performed using SPSS for Windows, Version 19.0 (SPSS Inc., Chicago, IL, USA).

## 5. Conclusions

The present study has provided insights into the molecular defense mechanisms of rice against *U. virens* in IR28. The resistance of cultivar IR28 was verified with two *U. virens* isolates via artificial inoculation. A distinct expression pattern was exhibited in IR28 that sustained immune response during *U. virens* infection compared to susceptible cultivar WX98 on a transcriptome level. CNLs and related pathways that were not suppressed were discovered when the first invasion of hyphae occurred at 5 dpi via GO enrichment and KEGG pathway analyses. Notably, the identification of candidate CNLs and their tissue-specific expression patterns in IR28 highlight their potential roles in the resistance mechanism. These findings significantly enhance our understanding of the rice–*U. virens* interaction, and future research should aim at the functional characterization of these candidate genes for the development of novel resistance resources in rice breeding. In the future, the function and contribution of candidate CNLs should be investigated and verified using gene editing and overexpression in transgenic plant construction. The mechanism of resistance of IR28 could be further explored by a deep investigation of the interaction between CNLs and plant defense pathways.

## Figures and Tables

**Figure 1 ijms-25-10655-f001:**
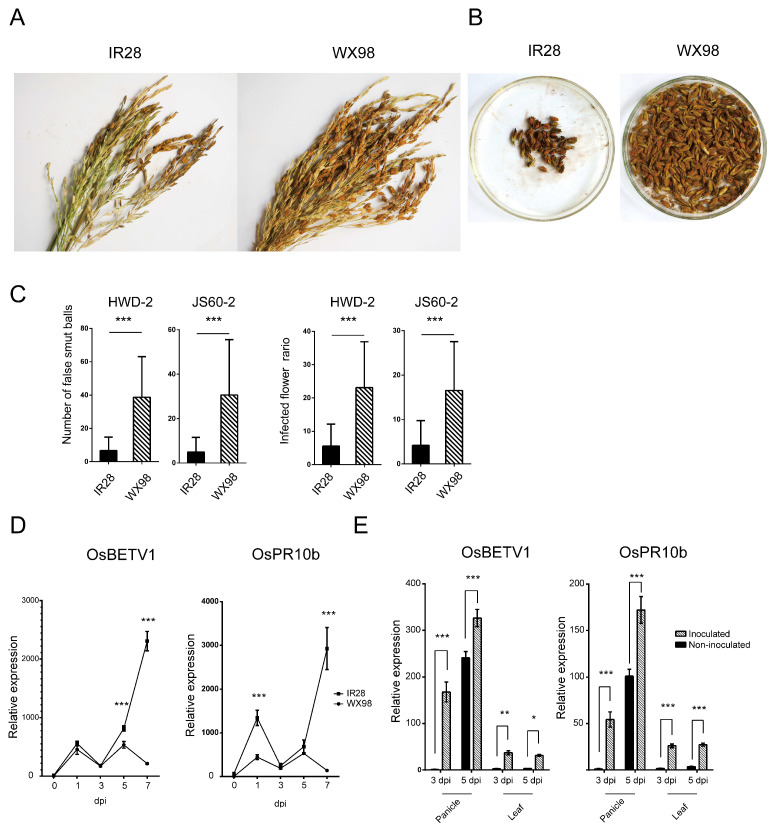
Cultivar IR28 was more resistant to rice false smut than cultivar WX98, and resistance-related genes were induced in IR28 panicles. (**A**,**B**): Three panicles infected by *U. virens* and their corresponding false smut balls on IR28 and WX98, respectively. (**C**): Number of smut balls and infected flower ratio. (**D**): Relative expression level of resistance-related genes OsBETV1 and OsPR10b in IR28 and WX98 under *U. virens* inoculation at 0, 1, 3, 5, and 7 dpi. Panicles were collected after artificial inoculation with *U. virens* HWD-2. Their RNAs were extracted using liquid nitrogen and reverse-transcribed into cDNA. Ubiquitin10 was set as reference gene. (**E**): Relative expression levels of OsBETV1 and OsPR10b in panicles, leaves, and stems at 3 and 5 dpi. *, **, and *** indicate significant difference between samples (*p* < 0.05, *p* < 0.01 and *p* < 0.001).

**Figure 2 ijms-25-10655-f002:**
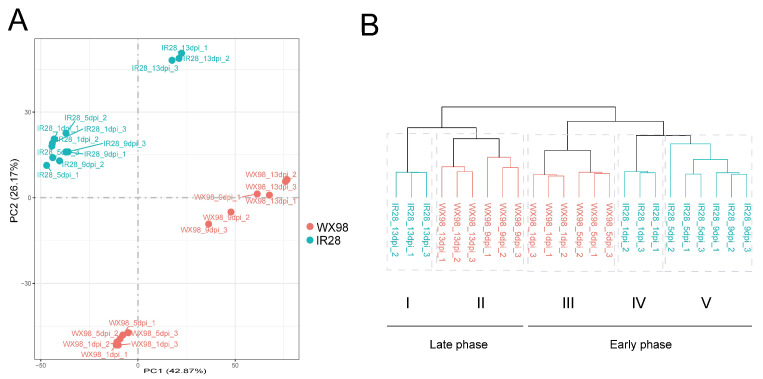
Principal component analysis (PCA) and hierarchical cluster analysis (HCA) revealed two phases of immune response during *U. virens* infection. (**A**): PCA of gene expression dynamics in resistant cultivar IR28 and the susceptible cultivar WX98 under *U. virens* infection. Four groups included samples from IR28 at 1, 5, and 9 dpi, IR28 at 13 dpi, WX98 at 1 and 5 dpi, and WX98 at 9 and 13 dpi, respectively. (**B**): HCA of gene expression dynamics in IR28 and WX98. Five subgroups included samples IR28 at 13 dpi, WX98 at 9 and 13 dpi, WX98 at 1 and 5 dpi, IR28 at 1 dpi, and IR28 at 5 and 9 dpi, respectively. Group I and II were clustered into late phase of infection. Group III, IV and V were clustered into early phase of infection.

**Figure 3 ijms-25-10655-f003:**
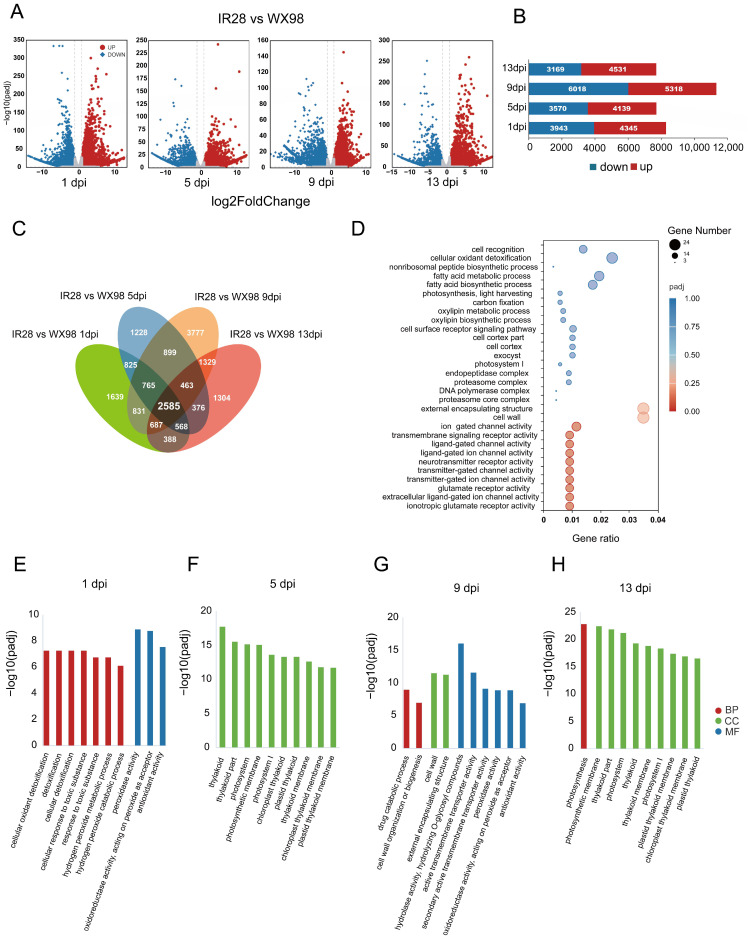
Differential gene expression analysis in IR28 compared to WX98. (**A**): Volcano plots display the DEGs identified in IR28 compared to WX98 during *U. virens* infection at 1, 5, 9, and 13 dpi. (**B**): Number of DEGs in IR28 at different dpi. (**C**): Venn diagram of the 2585 DEGs enriched at 1, 5, 9, and 13 dpi. (**D**): GO enrichment of DEGs in IR28 vs. WX98 at 1, 5, 9, and 13 dpi. (**E**–**H**): Ten most enriched GO terms of up-regulated DEGs in IR28 at 1 dpi, 5 dpi, 9 dpi, and 13 dpi, respectively.

**Figure 4 ijms-25-10655-f004:**
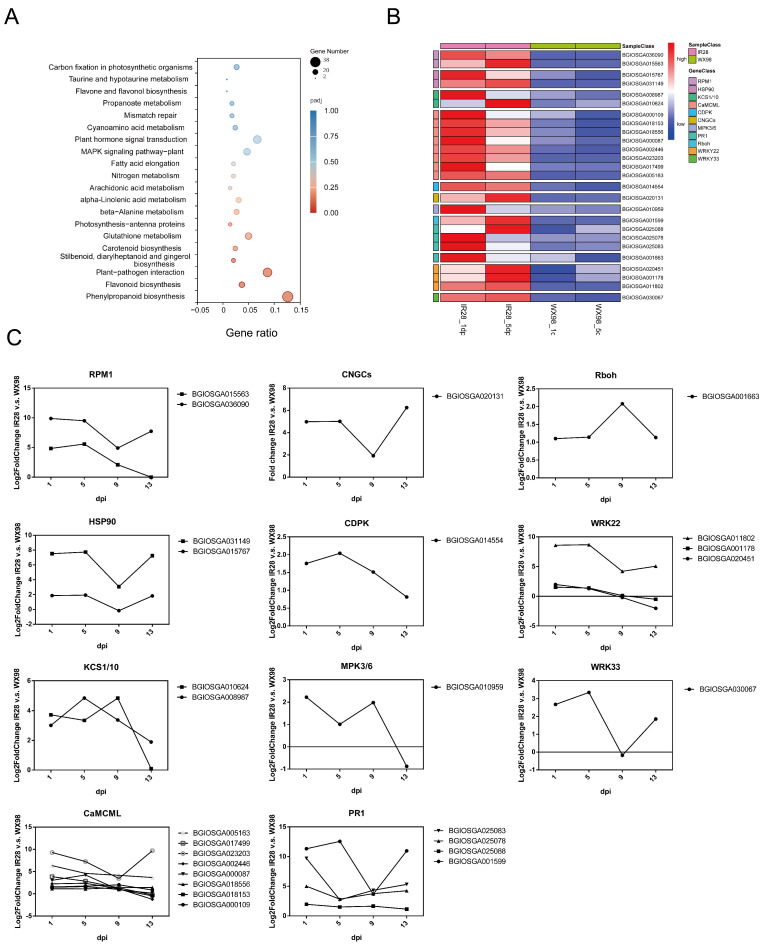
KEGG pathway enrichment analysis of DEGs in plant–pathogen interactions. (**A**): KEGG pathway enrichment analysis of a total of 2404 DEGs significantly up-regulated in IR28 compared to WX98 at both 1 and 5 dpi (adjusted *p*-value < 0.05). (**B**): Heatmap of DEGs enriched in the plant–pathogen interaction pathway. (**C**): Expression profile of DEGs enriched in nodes of plant–pathogen interaction pathway.

**Figure 5 ijms-25-10655-f005:**
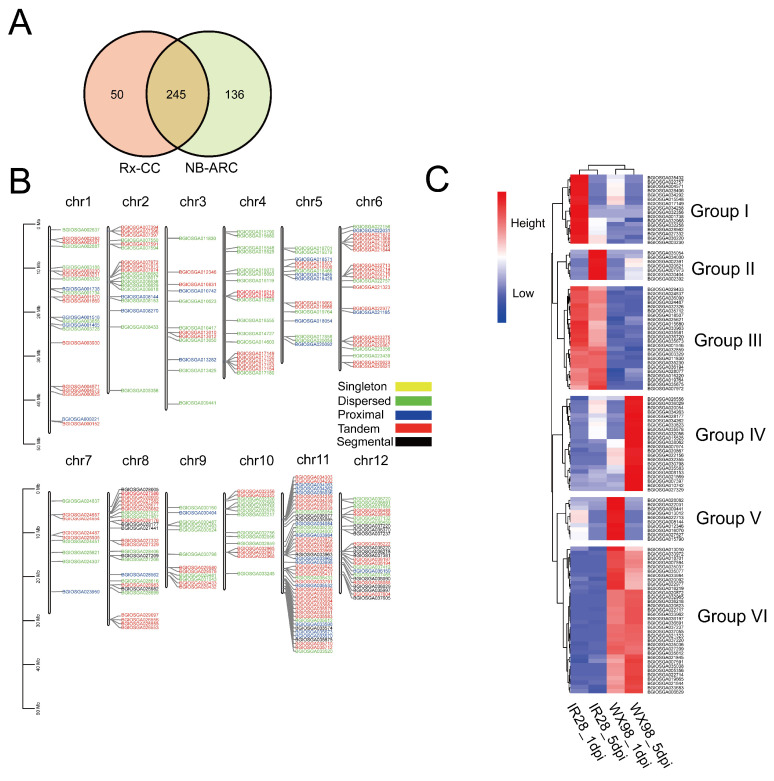
CNL identification in *Oryza sativa* Indica genome and expression profile. (**A**): Venn diagram exhibiting the number of proteins containing Rx-CC and NB-ARC domains. (**B**): CNL distribution in chromosomes and results of duplication analysis. (**C**): Hierarchically clustered heatmap of CNLs in IR28 and WX98 at 1 dpi and 5 dpi.

**Figure 6 ijms-25-10655-f006:**
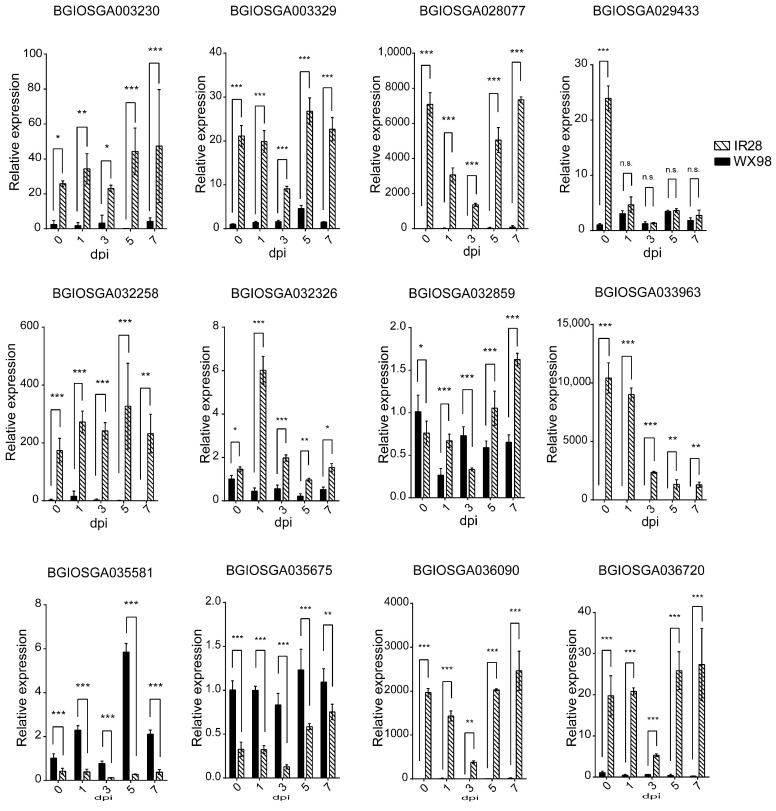
qRT-PCR validation of candidate CNL expression in IR28 and WX98 at 0, 1, 3, 5, and 7 dpi. *, **, and *** indicate significant difference between samples (*p* < 0.05, *p* < 0.01 and *p* < 0.001).

**Figure 7 ijms-25-10655-f007:**
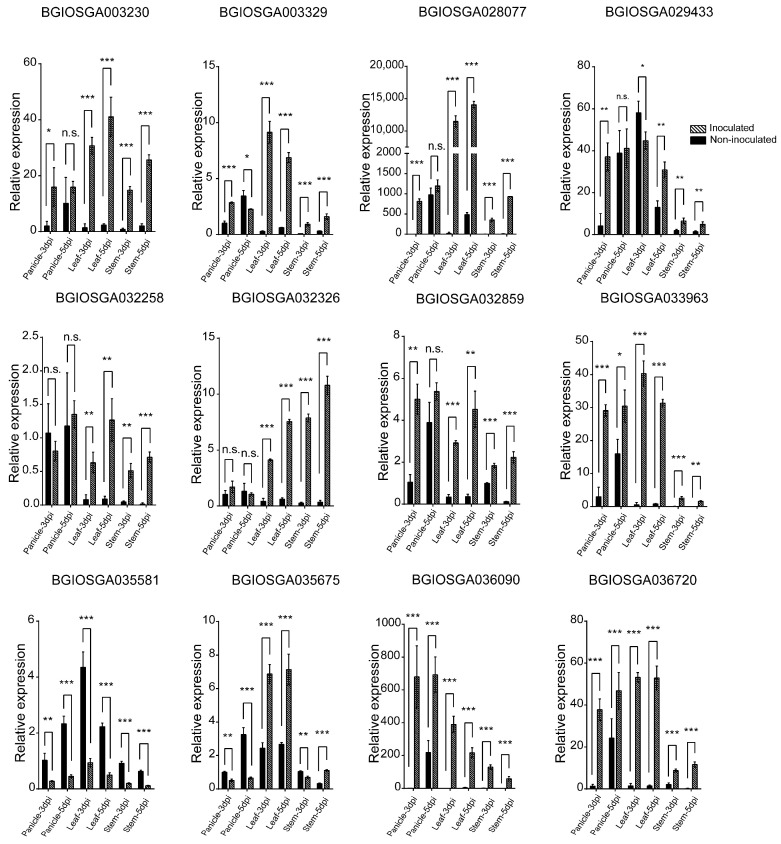
qRT-PCR validation of candidate CNL expression in IR28 during *U. virens* infection in panicle, leaves, and stems at 3 and 5 dpi. *, **, and *** indicate significant difference between samples (*p* < 0.05, *p* < 0.01 and *p* < 0.001).

**Table 1 ijms-25-10655-t001:** Susceptible analysis of rice cultivars IR28 and WX98 to *U. virens*.

	Number of Smut Balls		Infected Flower Ratio (%)	
	HWD-2	JS60-2	HWD-2	JS60-2
WX98	38.70 ± 24.40 ***	30.60 ± 24.96 ***	23.10 ± 13.76 ***	16.53 ± 10.98 **
IR28	6.65 ± 8.13	6.60 ± 6.31	5.54 ± 6.58	5.54 ± 5.33

**, and *** indicate significant difference between samples (*p* < 0.01 and *p* < 0.001).

**Table 2 ijms-25-10655-t002:** Up-regulated DEGs in KEGG pathway that were significantly enriched at 1 and 5 dpi.

KEGG ID	Pathway	Gene RATIO	*p*-Value	Adjusted *p*-Value	Count
osa00940	Phenylpropanoid biosynthesis	38/302	7.10 × 10^−8^	7.24 × 10^−6^	38
osa00941	Flavonoid biosynthesis	11/302	2.10 × 10^−5^	1.073 × 10^−3^	11
osa04626	Plant–pathogen interaction	26/302	4.03 × 10^−4^	1.37 × 10^−2^	26

**Table 3 ijms-25-10655-t003:** DEGs enriched in the KEGG pathway “Plant–pathogen interaction” (osa04626).

Gene IDs	Gene_Description	Domains	1 Day Post Inoculation		5 Days Post Inoculation	
			log2FoldChange	Adjusted *p*-Value	log2FoldChange	Adjusted *p*-Value
**CNGCs:**						
BGIOSGA020131	cyclic nucleotide-binding protein	CAP_ED	4.98	2.74 × 10^−21^	5.02	6.59 × 10^−23^
**CDPK:**						
BGIOSGA014554	calcium-dependent protein kinase	PKc_like	1.76	3.73 × 10^−13^	2.04	1.65 × 10^−11^
**Rboh:**						
BGIOSGA004406	NADPH_Ox and Efh protein	NADPH_Ox	0.65	2.48 × 10^−3^	1.13	4.46 × 10^−4^
BGIOSGA001663	Respiratory burst oxidase homolog protein B	NADPH_Ox, FNR_like, FAD_binding_1,	1.10	5.50 × 10^−19^	1.14	4.71 × 10^−6^
**CaMCML:**						
BGIOSGA000087	EF-hand	PTZ00183	3.05	1.95 × 10^−90^	4.24	4.88 × 10^−35^
BGIOSGA000109	EF-hand	PTZ00184	1.57	4.49 × 10^−31^	1.75	1.20 × 10^−18^
BGIOSGA002446	EF-hand	PTZ00184	2.26	2.86 × 10^−52^	2.38	4.50 × 10^−22^
BGIOSGA017499	EF-hand	PTZ00184	3.86	2.05 × 10^−61^	2.85	4.18 × 10^−9^
BGIOSGA018321	EF-hand	PTZ00184	0.13	9.02 × 10^−1^	1.32	1.294 × 10^−2^
BGIOSGA018556	EF-hand	PTZ00184	1.13	3.89 × 10^−20^	1.10	5.40 × 10^−6^
BGIOSGA018153	EF-hand	PTZ00184	1.50	2.97 × 10^−27^	1.63	1.02 × 10^−6^
BGIOSGA034766	EF-hand	Efh	0.98	2.42 × 10^−6^	1.00	9.81 × 10^−3^
**MPK3/6:**						
BGIOSGA010959	Mitogen-activated protein kinase	PKc_like	2.23	3.80 × 10^−56^	1.01	3.98 × 10^−5^
**WRKY33:**						
BGIOSGA030067	WRKY transcription factor	WRKY	2.68	1.67 × 10^−4^	3.35	2.29 × 10^−4^
**WRKY22:**						
BGIOSGA001178	WRKY transcription factor	WRKY	1.54	8.68 × 10^−7^	1.39	1.93 × 10^-4^
**PR1:**						
BGIOSGA001599	pathogenesis-related family 1 (PR-1)	CAP	11.32	7.16 × 10^−40^	12.56	3.64 × 10^−31^
BGIOSGA025083	pathogenesis-related family 1 (PR-1)	CAP	9.71	9.94 × 10^−15^	2.70	5.06 × 10^−7^
BGIOSGA025088	pathogenesis-related family 1 (PR-1)	CAP	1.95	8.07 × 10^−47^	1.51	4.71 × 10^−8^
BGIOSGA025078	pathogenesis-related family 1 (PR-1)	CAP	5.01	5.52 × 10^−112^	2.82	8.57 × 10^−5^
**RPM1:**						
BGIOSGA036090	NB-LRR disease-resistance protein	RX-CC, NB-ARC, LRR	9.88	2.50 × 10^−15^	9.51	1.46 × 10^−13^
BGIOSGA015563	NB-LRR disease-resistance protein	RX-CC, NB-ARC	4.84	1.11 × 10^−2^	5.58	2.34 × 10^−3^
BGIOSGA030404	NB-LRR disease resistance	RX-CC, NB-ARC, LRR	0.40	2.70 × 10^−1^	1.94	1.49 × 10^−6^
**HSP90:**						
BGIOSGA031149		HSP90	7.51	3.15 × 10^−8^	7.74	1.50 × 10^−8^
**KCS1/10:**						
BGIOSGA011802		PLN02192	8.59	2.27 × 10^−11^	8.69	1.28 × 10^−10^
BGIOSGA023203		PLN02192	9.29	6.89 × 10^−14^	7.26	8.30 × 10^−15^
BGIOSGA010624	type III polyketide synthase	CHS	3.72	1.89 × 10^−2^	3.35	8.87 × 10^−7^

**Table 4 ijms-25-10655-t004:** Candidate genes.

Gene IDs	Chr No.	Gene Length	Subcellular Location	1 Day Post Inoculation		5 Days Post Inoculation	
				log2FoldChange	Adjusted *p*-Value	log2FoldChange	Adjusted *p*-Value
BGIOSGA003230	1	2916	cyto *	4.93	9.03 × 10^−27^	2.37	1.97 × 10^−7^
BGIOSGA003329	1	2400	cyto	5.26	6.44 × 10^−74^	5.14	6.66 × 10^−40^
BGIOSGA028077	8	3954	plas	6.56	2.15 × 10^−78^	1.87	2.36 × 10^−18^
BGIOSGA029433	9	2937	cyto	1.28	9.39 × 10^−9^	1.24	2.84 × 10^−5^
BGIOSGA032258	10	2415	plas	9.64	2.32 × 10^−14^	2.06	2.64 × 10^−5^
BGIOSGA032326	10	2187	nucl	2.35	8.25 × 10^−24^	2.32	3.26 × 10^−10^
BGIOSGA032859	10	1080	cyto	1.06	2.27 × 10^−6^	1.34	1.68 × 10^−5^
BGIOSGA033963	11	3246	nucl	11.19	7.76 × 10^−20^	10.36	5.71 × 10^−16^
BGIOSGA035581	11	2940	nucl	2.66	1.28 × 10^−57^	1.89	2.17 × 10^−17^
BGIOSGA035675	11	3912	chlo	2.05	3.07 × 10^−24^	2.11	2.17 × 10^−20^
BGIOSGA036090	12	3243	cyto	9.88	2.5 × 10^−15^	9.51	1.46 × 10^−13^
BGIOSGA036720	12	3993	mito	4.86	1.6 × 10^−83^	2.63	5.23 × 10^−20^

* Cyto: cytoplasmic, plas: integral membrane protein, nucl: nuclear, chlo: chloroplast, mito: mitochondria.

## Data Availability

All raw sequencing reads generated in this study have been deposited in the National Center for Biotechnology Information Sequence Read Archive (NCBI-SRA) database under the Bio Project accession number PRJNA1112547.

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
