# Peer review of "Transcriptomic Analysis of the CNL Gene Family in the Resistant Rice Cultivar IR28 in Response to Ustilaginoidea virens Infection"

_ijms, 2024, doi:10.3390/ijms251910655_

Round 1

Reviewer 1 Report

Comments and Suggestions for Authors

In this manuscript, the authors performed a transcriptomic analysis of the rice resistance cultivar IR28 and susceptible cultivar WX98 response to Ustilaginoidea virens infection. They found the longer expression patterns in early infection phase of IR28, consistent with the sustained resistance response in it, and suppression of immune pathways when the hyphae first invade stamen filaments at 5 dpi. Most importantly, they found that CNLs plays key roles in IR28 response to Ustilaginoidea virens infection. The findings revealed a new insight of the rice-U. virens interaction. The following are my suggestions/concerns that need to be addressed for improving the manuscript.

The manuscript was not carefully prepared and proofread. Such as all Figures have low resolution, especially the labels from Figure 3D to Figure 3G are completely unclear, the line numbers are chaotic and some words in Table 1 are bold, while others are not.

Figure1B: The legend displays the “infection rate”, but there are no related images.

Figure1C: Please describe how the samples for qRT-PCR were prepared.

Please show the reference or other evidence to explain why OsBETV1 is selected here.

Please show the accession number of OsBSTV1 and OsPR10b.

Please show the Venn diagram of the DEGs at 1, 5, 9, and 13 dpi and described it.

Line 217: Are all the DEGs associated with ion channel activities and neurotransmitter receptor functions up-regulated after U. virens infection?

Figure 4C and Table 3: GO and KEGG analysis showed that enriched terms at 1 and 9 dpi related to immune response, but at 5 and 13 dpi related to plant development. Please explain why only select DEGs that significantly up-regulated in IR28 compared to WX98 at both 1 dpi and 5 dpi, but not the EDGs at 9 dpi and 13 dpi. Moreover, please describe the conclusions after the results.

Figure 5C: Please show the transcriptome profiling date of the 113 CNL genes.

Line149 (in discussion section): “IR48” should be “WX98”.

Comments on the Quality of English Language

No comment.

Author Response

Responses to editor and reviewers

Dear Editor and Reviewers,

It is very grateful for the review of the manuscript. We have made revisions based on the reviewers' comments and refined the text with English Editing service provided by MDPI.

Reviewer 1

In this manuscript, the authors performed a transcriptomic analysis of the rice resistance cultivar IR28 and susceptible cultivar WX98 response to Ustilaginoidea virens infection. They found the longer expression patterns in early infection phase of IR28, consistent with the sustained resistance response in it, and suppression of immune pathways when the hyphae first invade stamen filaments at 5 dpi. Most importantly, they found that CNLs plays key roles in IR28 response to Ustilaginoidea virens infection. The findings revealed a new insight of the rice-U. virens interaction. The following are my suggestions/concerns that need to be addressed for improving the manuscript.

Response: Thank you very much for your review and consideration.

The manuscript was not carefully prepared and proofread. Such as all Figures have low resolution, especially the labels from Figure 3D to Figure 3G are completely unclear, the line numbers are chaotic and some words in Table 1 are bold, while others are not.

Response: Thank you very much for pointing those out. All figures were exported with a higher resolution of 600 dpi from AI, and the automatic compression function of Word was turned off to make sure the figures are clear. The font of labels in Table 1 was corrected.

Figure1B: The legend displays the “infection rate”, but there are no related images.

Response: Thank you very much. Sorry, we did not show infection rate data but the description was incidentally remained. Because it may not reflect the resistant level of cultivars and “Number of smut balls per panicle” was normally used for resistant level evaluation in the studies of U. virens. We tried to add the infected flower ratio to evaluate more accurately in this study. The description of “infection rate” was removed.

Reference:

  1. Chen X, Liu C, Wang H, et al. Ustilaginoidea virens‐secreted effector Uv1809 suppresses rice immunity by enhancing O s SRT 2‐mediated histone deacetylation[J]. Plant Biotechnology Journal, 2024, 22(1): 148-164.

  1. Duan Y, Wang Z, Fang Y, et al. A secreted fungal laccase targets the receptor kinase OsSRF3 to inhibit OsBAK1–OsSRF3-mediated immunity in rice[J]. Nature Communications, 2024, 15(1): 7891.

Figure1C: Please describe how the samples for qRT-PCR were prepared.

Response: Thank you very much. The description of “Panicles were collected after artificial inoculation with U. viren HWD-2. Their RNAs were extracted with liquid nitrogen and reverse-transcripted into cDNA. Ubiquitin10 was set as the reference gene” was added to legend of Figure 1.

Please show the reference or other evidence to explain why OsBETV1 is selected here.

Please show the accession number of OsBSTV1 and OsPR10b.

Response: Thank you very much for the suggestion. The description of “OsBETV1 (LOC_Os12g36850) and OsPR10b (AAF85973.1) were used to evaluate resistance against U. virens in some studies [10,32]” was added to the methods.

Reference:

  1. Li, G.B.; He, J.X.; Wu, J.L.; Wang, H.; Zhang, X.; Liu, J.; Hu, X.H.; Zhu, Y.; Shen, S.; Bai, Y.F. Overproduction of osrack1a, an effector-targeted scaffold protein promoting osrbohb-mediated ros production, confers rice floral resistance to false smut disease without yield penalty. Molecular Plant 2022, 15, 1790-1806.
  2. Li, G.-B.; Liu, J.; He, J.-X.; Li, G.-M.; Zhao, Y.-D.; Liu, X.-L.; Hu, X.-H.; Zhang, X.; Wu, J.-L.; Shen, S., et al. Rice false smut virulence protein subverts host chitin perception and signaling at lemma and palea for floral infection. The Plant cell 2024, 36, 2000-2020.

Please show the Venn diagram of the DEGs at 1, 5, 9, and 13 dpi and described it.

Response: Thank you very much for the valuable suggestion. The Venn diagram was added to the Figure 3C and the description of “C: Venn diagram of the revealed 2585 DEGs enriched at 1, 5, 9, and 13 dpi.” was added to the legend of Figure 3.

Line 217: Are all the DEGs associated with ion channel activities and neurotransmitter receptor functions up-regulated after U. virens infection?

Response: Thank you very much for questioning the point. Part of them were up-regulated which is displayed in Table 3 according to KEGG enrichment analysis. They may play a role in the resistance against U. virens, but to be honest, we haven’t done work on them. The description at line 217 was modified to “The enrichment of these GO terms suggests that a variety of ion channel activities and neurotransmitter receptor functions are up-regulated in IR28, indicating a potent immune signal transportation occurs in IR28”

Figure 4C and Table 3: GO and KEGG analysis showed that enriched terms at 1 and 9 dpi related to immune response, but at 5 and 13 dpi related to plant development. Please explain why only select DEGs that significantly up-regulated in IR28 compared to WX98 at both 1 dpi and 5 dpi, but not the EDGs at 9 dpi and 13 dpi. Moreover, please describe the conclusions after the results.

Response: Thank you very much for pointing this out. Because infection of U. virens in WX98’s spikelets stopped pollen maturity and grain filling at 9 dpi, hypha was growing and embracing all the inner floral parts at 13 dpi according to the previous study. Thus, the receptor recognition of patterns or effectors of U. virens may have ended or worthless.

Reference:

“Fan, J.; Liu, J.; Gong, Z.Y.; Xu, P.Z.; Hu, X.H.; Wu, J.L.; Li, G.B.; Yang, J.; Wang, Y.Q.; Zhou, Y.F., et al. The false smut pathogen Ustilaginoidea virens requires rice stamens for false smut ball formation. Environmental Microbiology 2020, 22, 646-659.”

To avoid misunderstanding the description was modified to “At 9 dpi, the enrichment profile included pathways involved in “Phenylpropanoid bio-synthesis” (osa00940), “Pentose and glucuronate interconversions” (osa00040), and “alpha-Linolenic acid metabolism” (osa00592), which are consistent with results of on study in where infection of U. virens in WX98’s spikelets stopped pollen maturity and grain filling [1] Except for these grain-filling-related pathways, continued enrichment in the “Plant–pathogen interaction pathway” (osa04626) suggests a sustained host–pathogen interaction in IR28. At 13 dpi, the up-regulated DEGs were significantly enriched in pathways related to energy production and photosynthesis, such as “Photosynthesis-antenna proteins” (osa00196) and “Photosynthesis” (osa00195), indicating the interaction between U. virens and the plants has ended, consistent with hypha growing and embracing all the inner floral parts at 13 dpi [27].”

Figure 5C: Please show the transcriptome profiling date of the 113 CNL genes.

Response: Thank you very much for your precious suggestion. The transcriptome profiling date of 113 CNLs genes was added to the supplement file.

Line149 (in discussion section): “IR48” should be “WX98”.

Response: Thank you very much for pointing out the mistake. It was corrected to “WX98”.

Reviewer 2 Report

Comments and Suggestions for Authors

The reviewed manuscript contains quite interesting research results. I would like to ask the Authors to identify further research directions. What issues need to be investigated in the coming years? How can the research results obtained be used in the breeding of new varieties and in broad agricultural practice?  

The manuscript needs to be supplemented with conclusions.

Comments

Materials and Methods

Where was the research carried out? In which year?

Please give a brief description of the rice varieties studied.

References, please leave publications published no more than 10 years ago.

Author Response

Responses to editor and reviewers

Dear Editor and Reviewers,

It is very grateful for the review of the manuscript. We have made revisions based on the reviewers' comments and refined the text with English Editing service provided by MDPI.

Reviewer 2:

The reviewed manuscript contains quite interesting research results. I would like to ask the Authors to identify further research directions. What issues need to be investigated in the coming years? How can the research results obtained be used in the breeding of new varieties and in broad agricultural practice?

The manuscript needs to be supplemented with conclusions.

Response: Thank you very much for your valuable suggestion. To sum up and underline the importance the following description was added to the last paragraph. “In the future, the function and contribution of candidate CNLs should be investigated and verified using gene editing and overexpression in transgenic plant construction. The mechanism of resistance of IR28 could be further explored by a deep investigation of the interaction between CNLs and plant defense pathways.”

Comments

Materials and Methods

Where was the research carried out? In which year?

Response: Thank you very much for pointing this out. The following description was added to the methods “The inoculation experiments were conducted in a greenhouse at a temperature of 28°C with a humidity level exceeding 90%.” and “The pathogenicity analysis was performed in 2020 and 2021.”

Please give a brief description of the rice varieties studied.

Response: Thank you very much for the advice. The following description was added to the methods. “IR28 was cultivated by the International Rice Research Institute, with the cross of IR833-6-1-1-1/IR1561-149-1 as the maternal parent and IR1737 as the paternal parent. WX98 was developed by the Hubei Academy of Agricultural Sciences, with "D0424S" as the female parent and "Minghui 63" as the male parent. Both varieties are of the indica rice type.”

References, please leave publications published no more than 10 years ago.

Response: Thank you very much for your valuable suggestion. Citations were updated.

Reviewer 3 Report

Comments and Suggestions for Authors

Dear Authors,

Introduction part is efficient for the reader to analyzed Author’s obtained results, but I encourage to add the precise aim of studies, because we have right away, what kind of “analysis was performed”;

·       “Principal component analysis and Hierarchical cluster” why the capitals letter we have in the middle of the sentence?

·       Figure 1 should be enlarged because the valuable details are almost lost for the reader- especially panel A; Moreover, the differences between infected plants phenotype are unreadable in current form; what does it mean CK in figure 1 caption?

·       The main question is: what kind of confirmation Authors have for resistance as well as susceptible cultivar [IR28 and WX98]?  – Moreover, how to define the infection rate in figure 1panel B?

·       Analyses PR gene as a some kind of resistance potential is clear for plant pathologist, but please, describe why Authors use BETV1 in rice?

·       I warmly encourage English correction, because there are some places we have statements difficult to understand;

·       Materials and methods are quite good described; There are a lot of qRT-PCR data/analyses - What kind of two reference genes did Authors use to evaluate the relative gene expression? Because in supplementary files I find only ubiquitin gene?

·       Considering the numbers of obtained results the discussion part is very weak; Therefore, I suggest to strengthen this part also with deeply concluded part– maybe the solution will be adding also future prospects coming from obtained results underlining the importance of transcriptomic results for Oryza sativa cvs. “IR28” and “Wanxian98 (WX98)”-  U. virens strains interactions.

Comments on the Quality of English Language

I warmly encourage English correction, because there are some places we have statements difficult to understand;

Author Response

Responses to editor and reviewers

Dear Editor and Reviewers,

It is very grateful for the review of the manuscript. We have made revisions based on the reviewers' comments and refined the text with English Editing service provided by MDPI.

Reviewer 3:

Introduction part is efficient for the reader to analyzed Author’s obtained results, but I encourage to add the precise aim of studies, because we have right away, what kind of “analysis was performed”;

Response: Thank you very much for your valuable suggestion. The introduction was modified as your suggestion and the precise aim of the study was added at the beginning of the last paragraph of the introduction. “The aim of this study was to unveil resistant mechanisms and find resistant-related gene resources from the resistant cultivar IR28”

“Principal component analysis and Hierarchical cluster” why the capitals letter we have in the middle of the sentence?

Response: Thank you very much for reminder. The mistake of capitals letter has been corrected.

Figure 1 should be enlarged because the valuable details are almost lost for the reader- especially panel A; Moreover, the differences between infected plants phenotype are unreadable in current form; what does it mean CK in figure 1 caption? The main question is: what kind of confirmation Authors have for resistance as well as susceptible cultivar [IR28 and WX98]? – Moreover, how to define the infection rate in figure 1panel B?

Response: Thank you very much for pointing out details that need to be improved. The layout of Figure 1 was optimized, and panel A was enlarged and divided into two panels. All figures were exported with a higher resolution of 600 dpi from AI, and the automatic compression function of Word was turned off to make sure the figures were clear.

“CK” was substituted to “Non-inoculated” to describe more precisely.

Sorry, we did not show infection rate data but the description was incidentally remained. Because it may not reflect the resistant level of cultivars and “Number of smut balls per panicle” was normally used for resistant level evaluation in the studies of U. virens. We tried to add the infected flower ratio to evaluate more accurately in this study. The description of “infection rate” was removed.

Reference:

  1. Chen X, Liu C, Wang H, et al. Ustilaginoidea virens‐secreted effector Uv1809 suppresses rice immunity by enhancing O s SRT 2‐mediated histone deacetylation[J]. Plant Biotechnology Journal, 2024, 22(1): 148-164.

  1. Duan Y, Wang Z, Fang Y, et al. A secreted fungal laccase targets the receptor kinase OsSRF3 to inhibit OsBAK1–OsSRF3-mediated immunity in rice[J]. Nature Communications, 2024, 15(1): 7891.

Analyses PR gene as a some kind of resistance potential is clear for plant pathologist, but please, describe why Authors use BETV1 in rice?

Response: Thank you very much for pointing this out. Fan, et.al., 2019 first described OsBETV1 was induced up-regulated across U. virens infection. And some recent papers also use OsBETV1 for U. virens resistance evaluation.

Reference:

  1. Fan J, Du N, Li L, et al. A core effector UV_1261 promotes Ustilaginoidea virens infection via spatiotemporally suppressing plant defense[J]. Phytopathology Research, 2019, 1: 1-12.
  2. Li, G.B.; He, J.X.; Wu, J.L.; Wang, H.; Zhang, X.; Liu, J.; Hu, X.H.; Zhu, Y.; Shen, S.; Bai, Y.F. Overproduction of osrack1a, an effector-targeted scaffold protein promoting osrbohb-mediated ros production, confers rice floral resistance to false smut disease without yield penalty. Molecular Plant 2022, 15, 1790-1806.
  3. Li, G.-B.; Liu, J.; He, J.-X.; Li, G.-M.; Zhao, Y.-D.; Liu, X.-L.; Hu, X.-H.; Zhang, X.; Wu, J.-L.; Shen, S., et al. Rice false smut virulence protein subverts host chitin perception and signaling at lemma and palea for floral infection. The Plant cell 2024, 36, 2000-2020.

  • I warmly encourage English correction, because there are some places we have statements difficult to understand;

Response: Thank you for your precious suggestion. The manuscript was submitted to professional English Language Editing Services of MDPI.

  • Materials and methods are quite good described; There are a lot of qRT-PCR data/analyses - What kind of two reference genes did Authors use to evaluate the relative gene expression? Because in supplementary files I find only ubiquitin gene?

Response: Thank you very much for questioning. Sorry, we did use the ubiquitin gene as a reference gene according to recent references working on U. virens. OsUbi was used for qRT-PCR for rice genes in these papers.

Reference:

  1. Li, G.B.; He, J.X.; Wu, J.L.; Wang, H.; Zhang, X.; Liu, J.; Hu, X.H.; Zhu, Y.; Shen, S.; Bai, Y.F. Overproduction of osrack1a, an effector-targeted scaffold protein promoting osrbohb-mediated ros production, confers rice floral resistance to false smut disease without yield penalty. Molecular Plant 2022, 15, 1790-1806.

  1. Li, G.-B.; Liu, J.; He, J.-X.; Li, G.-M.; Zhao, Y.-D.; Liu, X.-L.; Hu, X.-H.; Zhang, X.; Wu, J.-L.; Shen, S., et al. Rice false smut virulence protein subverts host chitin perception and signaling at lemma and palea for floral infection. The Plant cell 2024, 36, 2000-2020.

  1. Fan J, Du N, Li L, et al. A core effector UV_1261 promotes Ustilaginoidea virens infection via spatiotemporally suppressing plant defense[J]. Phytopathology Research, 2019, 1: 1-12.

  • Considering the numbers of obtained results the discussion part is very weak; Therefore, I suggest to strengthen this part also with deeply concluded part– maybe the solution will be adding also future prospects coming from obtained results underlining the importance of transcriptomic results for Oryza sativa cvs. “IR28” and “Wanxian98 (WX98)”-  U. virens strains interactions.

Response: Thank you very much for the great advises.

To deeply discuss resistance-related pathways the following description was added to the third paragraph of discussion. “The results of the GO enrichment analysis and KEGG pathway analysis provide strong evidence that IR28 is more resistant than WX98. DEGs significantly enriched in plant-defense-signal-related GO terms in IR28 confirmed the resistance to U. virens at the transcriptomic level. At the same time, the fact that DEGs were enriched in panicle development and grain-filling-related pathways also revealed those processes were intercepted by U. virens in WX98 at 5, 9, and 13 dpi at the KEGG pathway level.”

To underline the importance the following description was added to the last paragraph. “In the future, the function and contribution of candidate CNLs should be investigated and verified using gene editing and overexpression in transgenic plant construction. The mechanism of resistance of IR28 could be further explored by a deep investigation of the interaction between CNLs and plant defense pathways.”

Round 2

Reviewer 3 Report

Comments and Suggestions for Authors

In my opinion Authors siginificantly improved the manuscript quality- espacially, in figures quality as well as in discussion part with  added some future prospects. Moreover, the Authors expalnation are logical and comprehensive. Furthermore, English quality is definitely better in current form.

On the other side, the weakness of the current form version is the lack of tracking changes" -difficult for reviewer work.